

# Comparison of cavity enhanced optical–feedback laser spectroscopy and gas chromatography for ground-based and airborne measurements of atmospheric CO concentration

Irène Ventrillard[1], Irène Xueref-Remy[2], Martina Schmidt[2,3], Camille Yver Kwok[2], Xavier Faïn[4], and Daniele Romanini[1]

[1]Univ. Grenoble Alpes, CNRS-UMR5588, Laboratoire Interdisciplianire de Physique (LIPhy), Grenoble, France
[2]Laboratoire des Sciences du Climat et de l'Environnement (LSCE), UMR CEA-CNRS 1572,Gif-sur-Yvette, France
[3]Now at Institut für Umweltphysik (IUP) University Heidelberg, Germany
[4]Univ. Grenoble Alpes, Institut des Géosciences de l'Environnement (IGE), Grenoble, France

*Correspondence to:* Irène Ventrillard (irene.ventrillard@univ-grenoble-alpes.fr)

**Abstract.**

We present the first comparison of carbon monoxide (CO) measurements performed with a portable laser spectrometer that exploits the Optical–Feedback Cavity–Enhanced Absorption Spectroscopy (OF-CEAS) technique, against a high performance automated gas chromatograph (GC) with mercuric oxide reduction gas detector. First, measurements of atmospheric CO mole
fraction were continuously collected in Paris (France) suburb over one week. Both instruments showed an excellent agreement within typically 2 ppb (part per billion in volume) fulfilling the World Meteorological Organization (WMO) recommendation for CO inter-laboratory comparison. The compact size and robustness of the OF-CEAS instrument allowed its operation aboard a small aircraft employed for routine tropospheric air analysis over the French Orléans forest area. Direct OF-CEAS real–time CO measurements in tropospheric air were then compared with later analysis of flask samples by the gas chromatograph.
Again, a very good agreement was observed. This work establishes that the OF-CEAS laser spectrometer can run unattended at a very high level of sensitivity (<1 ppb) and stability without any periodic calibration.

## 1 Introduction

Carbon monoxide (CO) is a reactive trace gas that plays a significant role in global atmospheric chemistry by being a major sink of tropospheric hydroxyl radicals (OH). Hydroxyl radical is the main tropospheric oxidant, thus its abundance affects the
lifetimes of radiatively important gases such as methane. Oxidation of CO by OH also provides a source or a sink, respectively in high or low NOx conditions, for tropospheric ozone (Logan et al., 1981). CO concentration in the atmosphere have thus crucial implications for both climate and air quality issues, and accurate CO measurements in the troposphere are important when modeling climate-chemistry interactions with global coupled models (Voulgarakis et al., 2013).

Consequently, monitoring of tropospheric CO has been conducted over the last decades on the global scale (Novelli et al.,
1998). Recently satellite-based observations have become an important contribution to regional monitoring of atmospheric CO (Worden et al., 2013). There is still a need, however, to strengthen direct CO observations from both surface stations and



aircraft to assess the large spatio-temporal variability of CO, especially within the boundary layer at the regional scale for a better understanding of atmospheric chemistry and transport, and towards improving forecast modeling of air quality (Sahu et al., 2013; Warner et al., 2013; Té et al., 2016).

Historical methods, such a gas chromatography, have been used for many years for surface monitoring of CO (Derwent et al.,
2001; Langenfelds et al., 2002; Yver et al., 2009; Schmidt et al., 2014). A gas chromatograph (GC) equipped with mercuric oxide reduction gas detector allows for very sensitive laboratory measurements but requires hourly calibration procedures with calibration gases and an expert operator to achieve uniform high–quality results. In addition, the mercuric oxide reduction detectors are known for their non-linear response function, which needs to be quantified on a regular basis several times per year (Yver et al., 2009). However, recent developments in optical spectroscopy methods have brought new alternatives
for in-situ CO monitoring (Zellweger et al., 2012; Chen et al., 2013; Yver Kwok et al., 2015). The most sensitive optical techniques allows detection limit at the ppb level. Among them, Optical–Feedback Cavity Enhanced Absorption Spectroscopy (OF-CEAS) (Morville et al., 2005) exploits a high finesse optical cavity in which is coupled a laser source to enhance the interaction of photons with gas molecules present inside the cavity (Morville et al., 2014). OF-CEAS based measurement of CO concentration has been conducted before around various applications, for example for in-situ trace measurements on
geothermal gases (Kassi et al., 2006), for continuous and high resolution measurement of air extracted from ice cores drilled out of polar glaciers (Faïn et al., 2014) or for breath analysis in different medical settings (Ventrillard-Courtillot et al., 2009; Maignan et al., 2014).

In this study, we report on the comparison for both surface and airborne CO measurements by OF-CEAS and by a high performance gas chromatograph equipped with a mercuric oxide reduction gas detector. The atmospheric CO concentration
in Gif-sur-Yvette, France, was continuously analysed at ground level over one week. Then, the OF-CEAS instrument was set aboard a small aircraft employed for periodic tropospheric air measurements over the French Orléans forest area. Airborne in-situ CO measurements by OF-CEAS were then compared with flask samples later analysed with the GC at LSCE. With this comparison we demonstrate that OF-CEAS can become a work-horse in many CO applications for environmental (including atmospheric) applications, which demand robust and compact instrumentation with ppb sensitivity and a response time faster
than 1 s.

All values reported in this paper are dry air mole fractions (expressed in ppm or ppb) but are called concentration as commonly done by the community.

## 2    Materials and Methods

We briefly describe the GC set-up and outline the OF-CEAS technique, highlighting the characteristics most relevant for the
measurements reported here such as instruments calibrations. In particular two steps of post data processing were needed to come to an excellent agreement between the optical and chromatographic measurements performed during autumn 2006. Firstly, the non-linearity of the GC reduction gas detector was corrected following a procedure established in 2010. Secondly,





the two instruments had to be calibrated on the same standard scale (WMO CO X2004). This was performed with a recent re-evaluation (in 2014) on this scale of the gas standards initially used for the OF-CEAS spectrometer calibration.

## 2.1 Gas chromatograph

The LSCE laboratory at Gif-sur-Yvette is equipped with two coupled gas chromatographs (HP-6890, Agilent and PP1, Peak laboratories) which run fully automated, alternating between calibration gas and ambient air, in order to analyse CO, $H_2$, $CO_2$, $CH_4$, $N_2O$ and $SF_6$ concentration in atmospheric measurements, flask samples or high pressure cylinders. Detailed descriptions of the GC system for CO analysis is given by Yver et al. (2009). CO is analysed with the PP1 chromatograph equipped with a reduction gas detector (RGD) after reduction of mercuric oxide and detection of mercury vapor by UV absorption. Each analysis takes less than 6 min allowing between two and six injections of ambient air alternating with calibration gases and flask samples. The air is dried before injection during two steps. First, it passes through a glass trap which is hosted in a commercial refrigerator kept at $5^o$ C in order to remove a large fraction of water vapor and in a second step, air is further dried by passing through a second glass trap cooled in an ethanol bath at $-55^o$ C using a cryogenic cooler. An operator is only required to change the cooling trap 2-3 times per week and to restart the acquisition.

**Calibration of the gas chromatograph: Correction of the reduction gas detector nonlinearity**

The GC is calibrated for CO with cylinders certified by NOAA/GMD on the WMO CO X2004 scale (Novelli et al., 1994). CO concentrations are calculated using regular measurements of one calibration cylinder with a typical atmospheric concentration value (here $168.0 \pm 0.8$ ppb) and a non linear correction function of the detector response as described in Yver et al. (2009) and Yver (2010). The correction function is determined on an annual frequency using a set of 5 cylinders with a CO concentration ranges from $57 \pm 1.0$ ppb to $523$ ppb$\pm 10.9$ ppb, and applied as a post-run correction (Yver et al., 2009). This non linear correction was validated using flask measurement comparisons between LSCE and NOAA, with a mean difference of $4.5 \pm 2.2$ ppb for the period of July 2006 to July 2009. For the one week comparison campaign with the OF-CEAS instrument in November 2006, the correction function applied to CO in-situ measurements by the GC ($CO_{meas}$) to obtained the calibrated CO concentrations reported in the following is given by :

$$\Delta CO_{corr} = 11.4 + 0.077 \times CO_{meas} - 7.1\ 10^{-4} \times (CO_{meas})^2 + 1.03\ 10^{-8} \times (CO_{meas})^3 \qquad (1)$$

It applies a correction for the non-linear behavior of the analyser in the range of - 15 to + 15 ppb for measured CO concentration up to 500 ppb.

The calibration cylinder is analysed every 30-40 minutes as well as a quality control gas, a so called target gas, with a CO concentration of 68 ppb, that is treated as unknown. Over the entire comparison period, the repeatability defined as 1-sigma standard deviation of the target gas is 0.4 ppb.

The flask samples filled during the airborne campaign are measured in a similar way as the ambient air concentration with two injections per flask.





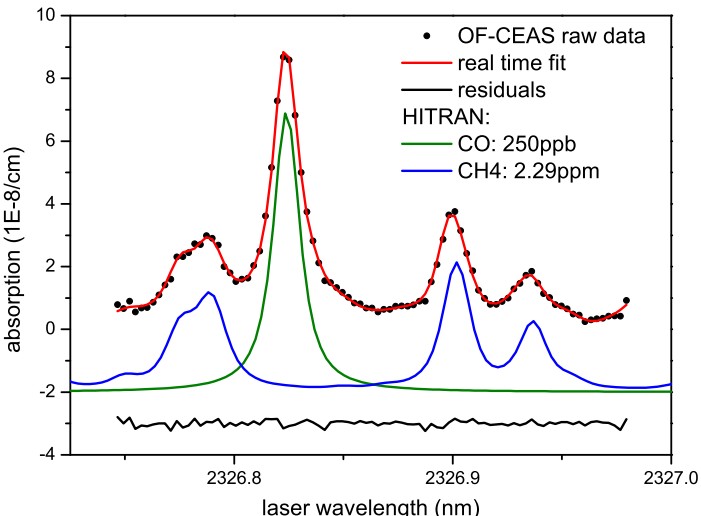

**Figure 1.** Single OF-CEAS spectra in absolute absorption units, recorded in 150 ms with a gas sample at a pressure of 200 mbar and a temperature of 295 K. CO and $CH_4$ concentrations are deduced from the real time fit. The standard deviation of the residuals is $9 \times 10^{-10}$ cm$^{-1}$ (in absorption units). The base line has been subtracted for comparison with HITRAN simulated absorption spectra (Rothman et al., 2013). HITRAN spectra and the residuals have been offset for clarity.

## 2.2 Optical-Feedback Cavity Enhanced Absorption Spectrometer

The laser spectroscopy technique going under the name of OF-CEAS was introduced by Morville et al. (2005) and has been further detailed in different publications (Kerstel et al., 2006; Kassi et al., 2006; Ventrillard-Courtillot et al., 2009; Faïn et al., 2014; Maignan et al., 2014; Morville et al., 2014). In particular the OF-CEAS instrument used in this study has been described

5  in Kassi et al. (2006). It provides in situ CO measurements with a detection limit of 0.2 ppb in 20 s (Faïn et al., 2014) with no calibration and running unattended. Here we will just recall the basic principle of OF-CEAS.

Spectroscopic measurements of trace gas concentrations require a long light absorption path. Like other spectroscopy techniques, OF-CEAS is based upon the use of a sample cell made with an optical cavity in order to enhance light interaction with the gas sample. Specifically in the spectrometer used here for CO monitoring, the resonant optical cavity composed of

10  high-reflective mirrors (mirror reflectivity: R≃99.995%) allows a 20 km effective absorption length with a compact set-up: the cavity length is only 1 m long, folded to 0.5 m external size. The complexity of using a resonant cavity with high reflectivity mirrors (thus very small transmissivity) resides in the coupling of a sufficient amount of light in the cavity by injecting laser light through one of the cavity mirrors. The originality of OF-CEAS is that the optical cavity is made of three mirrors placed in a "V-shaped" configuration. In this way, a fraction of the light trapped inside the optical cavity, and therefore frequency-selected





by a resonant mode of the cavity, can be returned to the laser. The non-linear response of the laser then forces it to lase on the exact frequency of the excited cavity "mode". This optical feedback (OF) effect is also responsible for a narrowing of the laser emission line width and an increase of the cavity transmission to a level that is orders of magnitude larger than in competing techniques (Morville et al., 2005). OF-CEAS absorption spectra are acquired on a small spectral region, as shown in Fig. 1 for

the present case, by scanning the laser frequency at a fast acquisition rate (6 Hz here). However the measurement response time is not limited by this rate but by the gas exchange rate inside the sample volume. Therefore, the cell is designed to allow minimal dead spaces and a small internal (sample) volume, which does not exceed 18 cm$^3$. In this study, the gas is continuously flowing at 250 sccm (standard cm cubes per minute) with a cell pressure stabilized to a relatively low value (200 mbar), providing a typical gas exchange time below 1 s, which can be easily improved if needed by using a lower sample pressure or

a higher pumping rate. The design of the spectrometer is robust and compact: the optical assembly and all the electronics, for real time control and data acquisition, fit inside a 19" chassis.

Importantly, OF-CEAS provides quantitative absorption measurements in real-time without the need for a periodic calibration with certified gas mixtures. A normalization procedure of the absorbance scale is realized continuously based on cavity optical loss measurements performed by Cavity Ring Down Spectroscopy (CRDS) (Kerstel et al., 2006; Morville et al., 2014).

Pressure stabilization inside the cavity allows for a real-time numerical fit of the measured absorption spectra with a reduced number of parameters (Gorrotxategi-Carbajo et al., 2013). This enables the selective determination of the concentrations of all compounds that possess absorption lines in the selected spectral window (CO and CH$_4$ here). This is a key point in trace gas monitoring, atmospheric air being a highly complex gas mixture. To optimize CO limit of detection (LOD), the laser emission is chosen in the near infrared region (NIR), in the [2.3- 2.4] $\mu$m range (Fig. 1), an interesting region that includes an

atmospheric window where water vapor absorption lines are sparse and weak, while several light species such as CO display relatively strong absorption bands. The smallest detectable absorption coefficient is typically in the range of several $10^{-10}$/cm. As a result, an OF-CEAS instrument optimized for CO monitoring has a LOD of 0.2 ppb of CO for an acquisition time of 20 s, value derived from the Allan variance in Faïn et al. (2014). At longer acquisition times, small drifts prevent for better averaging (Morville et al., 2014). These drifts however are small and remain bounded as shown by the Allan variance of CO

measurements in Faïn et al. (2014).

Fast response time and low LOD allow OF-CEAS instruments to perform fast trace gas monitoring. This has already been exploited in airborne atmospheric measurements -of methane in Romanini et al. (2006), water isotopes in Iannone et al. (2009a) and Kerstel et al. (2006), and CO in this work- as well as in other fields as mentioned in the introduction section. The performances of the OF-CEAS technique led a private company (AP2E, Aix-en-Provence, France) to exploit the patent. OF-CEAS

spectrometers are now commercialized (namely ProCEAS) in the domains of industrial and air quality monitoring, with some very stringent applications such as air quality control onboard nuclear submarines. ProCEAS work with lasers in the NIR region and standard ProCEAS optimized for CO detection reach a LOD of 1 ppb in 1 s. Recent technological progress has allowed other companies to develop instruments based on different laser spectroscopy techniques that reach a sub ppb LOD, such as Los Gatos by Off-Axis Integrated Cavity Output Spectroscopy (OA-ICOS) or Picarro by CRDS.




**Calibration of the OF-CEAS Spectrometer: Conversion of absolute molecular absorption to CO concentration value**

The ring-down calibration included in the OF-CEAS technique allows for direct absolute molecular absorption measurements (in $cm^{-1}$ unit, Fig. 1). Then line intensity is directly converted to CO concentration with a conversion factor specific to the fitted absorption line for the temperature and pressure operation conditions (298.5±1 K and 200±1.8 mbar in this work). This

allows to account for temperature and pressure effects on the line intensity parameters. It is important to stress that this factor is a constant that does not depend on the cavity finesse (continuously measured by ring-down) nor on the gas sample composition (the multiline fit allows independent fit of each species). The latest assertion implies that the foreign pressure broadening effect on CO absorption lines from water is neglected. This is justified for atmospheric measurements where water concentration remains small (it varies from 0.8% to 1.6% for ground based measurements reported in this work). As a consequence, the

conversion factor of each molecule needs to be determined for the OF-CEAS spectrometer working conditions only once and then during operation the instrument delivers absolute concentrations in real-time without the need of any calibration with certified mixture.

The conversion factor can be derived from spectral database or by direct calibration using certified mixture. Even if the line intensities for CO in this spectral region are well defined -to better than 1% in the HITRAN database (Li et al., 2015)-,

in practice calibration with gas standards is found to be more accurate because it cancels sensors pressure and temperature absolute accuracy and allows to minimize line profile effects by considering a specific model in the fit procedure -a Rautian model is used as in Gorrotxategi-Carbajo et al. (2013). As an example, the conversion factor computed only from the line intensity given in HITRAN would lead to a 10% overestimation of the concentration value in the present case.

For the comparison campaign in 2006, the OF-CEAS spectrometer was calibrated with 2 high pressure cylinders containing

air whose CO concentration had been certified in 1995 by the Commonwealth Scientific and Industrial Research Organisation (CSIRO). However, the GC was calibrated on the WMO X2004 scale provided by the NOAA. Differences between CSIRO and NOAA CO scales in the order of 6 ppb have been reported by Masarie et al. (2001). Therefore we reevaluated CO concentrations in the CSIRO cylinders against the WMO CO X2004 scale. This was done in 2014 using another OF-CEAS instrument, designed for ice core analysis (Faïn et al., 2014). This instrument was calibrated with three standards certified in 2011 by the

NOAA GMD (Global Monitoring Division) Carbon Cycle Group on the WMO CO X2004 scale ($33.2 \pm 0.5$ ppb, $51.8 \pm 0.1$ ppb and $102.1 \pm 0.1$ ppb of CO). The two working standards used to calibrate the instrument for the 2006 campaign, were recalibrated to $35.0 \pm 1.5$ ppb and $104.0 \pm 1.5$ ppb while the CSIRO values certified in 1995 were $32.6 \pm 0.7$ ppb and $98.7 \pm 1$ ppb, respectively.

These WMO-scaled CO standard gases values were then used to calibrate the entire 2006 dataset using a linear relation-

ship (i.e., without offset adjustment), which is consistent with the fact that the zero of spectral measurements is intrinsically accurate. The accuracy of this calibration is then estimated to be of 2 % limited by the accuracy of the NOAA standards. The reproducibility of the OF-CEAS spectrometer is much better, at the level of 0.2 ppb for an acquisition time of 20 s. Furthermore, the high linearity of OF-CEAS was previously reported for concentrations ranging over more than three decades (data published for water measurements in Iannone et al. (2009b)).





## 3   Comparison: Results and discussion

### 3.1   In-situ ground measurements

Direct comparison of atmospheric CO concentration measurements by GC and OF-CEAS over one week (8/11/2006 - 14/11/2006), was performed at LSCE in Gif-sur-Yvette, 20 km southwest of Paris (48°43′N / 02°09′E / 120 m above sea level). The LSCE

GC setup routinely monitors atmospheric concentration with a sampling inlet located on the roof of the building, 7 m above ground level. The OF-CEAS instrument from LIPhy was set to run in the same building but with an idenpendent sampling line. Sampling lines measured about 20 m and were made of 3/8" diameter Dekabon tubes. The estimated sample propagation delay along the tube from the roof to the OF-CEAS instrument is about 1 min (with a gas flow of 250 sccm). A larger delay of about 15 min is observed on the GC data mainly due to the use of the cold trap. The volume of this trap corresponds to the sample

volume collected over 15 min by the GC, inducing a smoothing of the signal of the semicontinuous injections. Reported GC CO values are dry air mole fractions. For the comparison, OF-CEAS data were expressed in the same way by accounting for the humidity rate routinely monitored simultaneously with atmospheric temperature and pressure at the Saclay tower (meteorological data provided by the SPR group from Saclay CEA). This tower is located about 1.5 km North-Northwest from the sampling point.

OF-CEAS and GC raw data were post-treated as detailed in section 2. In Fig. 2 are shown typical variations of atmospheric CO dry concentration measured by both the GC and the OF-CEAS analysers during a week day (2.a), a Sunday (2.b) and a Wednesday night (2.c). During night-time and most of the day on the week-ends, CO concentration slowly varies within typically 100-300 ppb. But during week days emissions from nearby traffic usually induce two rush hour peaks in the morning at around 8 am and in the evening after 5 pm. The persistence and higher intensity of the second CO peak could be related to

air mass change or to the Friday evening traffic jams all around Paris suburbs (Fig. 2.a). During daytime and in the evening, the fast response time of the OF-CEAS instrument (1 s averaged here to 2 s) allows to record many short but very strong peaks (sometime rising up to more than 1 ppm during only 1 or 2 minutes). These are due to local pollution of vehicles passing by close to the laboratory. The 14/11/2006 night (Fig. 2.c) when CO concentration remained around 100 ppb and very small concentration fluctuations are measured (less than 20 ppb over 7 hours), allows a comparison over several hours with nearly no

effect from the slower GC response time.

OF-CEAS and GC measurements show an excellent agreement. When CO concentration varies slowly, such as during night-time and Sunday measurements shown in Fig. 2, the agreement is within about 2 ppb rms over several hours for concentration values ranging from 100 ppb to 300 ppb for the whole comparison period of one week. This difference is fully compatible with the calibration accuracy of the two instruments reported before. When CO concentration is subject to fast changes such as

in Fig. 2.a, the strong difference in the GC and OF-CEAS measurements is explained by the slower response time of the GC instrument due to the buffering effect of the cooling trap. The air sample being continuously flushed in the cooling trap, the effect on CO concentration measurements by the GC is not equivalent to a simple time average. A more complex weighted moving averaging could be performed on the faster OF-CEAS measurement to try to mimic the GC measurement, but a study concerning this averaging issue appears to be beyond the scope of this paper.



The OF-CEAS instrument measures CO and $CH_4$ at the same time (Fig. 1). Contrary to CO, $CH_4$ concentration is not sensitive to traffic pollution. Daily variability is usually less than 10% with a background value of about 1900 ppb. The rms noise of the OF-CEAS measurements for $CH_4$ is 4 ppb for an averaging time of 20 s. A good agreement between OF-CEAS and GC is also found with maximum deviations of ± 20 ppb, corresponding to about 1% in relative unit. Such performance

has been previously reported in Romanini et al. (2006) with a similar OF-CEAS instrument compared to the same GC.

## 3.2 Airborne measurements

In the framework of the French RAMCES observation network for greenhouse gases monitoring, regular weekly flights have been carried out by LSCE since 1996 above the Orléans forest, located about 100 km south of Gif-sur-Yvette. This flight program aims at improving our understanding of transport processes into the atmospheric boundary layer, and to better assess

the relative role of local, regional and continental anthropogenic and biospheric fluxes on the observed trace gas concentrations. Especially, vertical profiles of trace gases are very useful for assessing atmospheric transport model performances. During the flights, air samples are collected in flasks, as described in Chevalier et al. (2009), and later analysed at LSCE by GC to measure the concentration of $CO_2$, $CH_4$, CO, $N_2O$ and $SF_6$ (Xueref-Remy et al., 2011; Haszpra et al., 2012). Glass flasks are filled at 10 different levels (between 100 m and 3000 m above ground level). Those used for the present comparison were analyzed

one week after collection. On the 15/11/2006, the OF-CEAS instrument was installed inside the aircraft to measure in situ CO atmospheric concentration during the entire flight. The OF-CEAS instrument was mounted in a 19″ rack fixed in place of a seat.

Tropospheric air was sampled upwind of the aircraft engines exhaust: a 2 m Dekabon inlet line carries outside air to the set-up entrance, passing through a customized window of the aircraft. The same inlet was used for the OF-CEAS instrument.

For the GC, the sampling unit consists of a diaphragm pump which draws air through the chemical drying cartridge filled with $Mg(ClO_4)_2$. Air is collected in 1 L glass flasks sealed with PTFE O-rings. Flasks are collected in pairs and pressurized to 2 bar absolute pressure. The filling step takes between 30 s and 1 min during which the plane covers a typical distance of 5 km.

The entire set of measurements is shown in Fig. 3: starting from the airport of Toussus-le-Noble ($48^o45'$N/ $2^o08'$E/ 164 m asl), during the flight to the Orléans forest area ($47^o50'$N/ $2^o30'$E/ 135 m asl) where the plane starts a routine flight that consists

of legs at pre-defined altitudes for the flask samples collection, and during flying back to the airport. During the flight above the Orléans forest, CO levels remain around 90 ppb (± 10 ppb) above 1000 m, while an increase at lower altitude is clearly measured up to 150 ppb at 100 m due to soilborne CO sources like traffic and heating.

It should be highlighted that the OF-CEAS instrument is robust enough to operate in the harsh environment of a small aircraft including during take-off and landing phases. On the tarmac, CO rises up to 16 ppm due to airplane exhaust gases.

This illustrates the wide sensitivity range of the measurements, of about 4 orders of magnitude. During the whole flight, the instrument ran un-attended. The only not automatized action by the operator required during the flight consisted in adjusting a needle valve at the inlet of the sampling cell specifically plumbed to maintain a constant flux to 50 sccm while ambient pressure was subject to change with altitude. Later, this process has been automatized using a numerically controlled flux regulator. OF-CEAS data are averaged for 2 s, time that corresponds to a space resolution of only 100 m according to the aircraft velocity.





Due to the harsh environment in the plane, the standard deviation of the measurements is increased to typically 2 ppb (zoom in Fig. 3) while for ground-based measurements it is 0.6 ppb for the same 2 s averaging time (Fig. 2.c).

As explained in the previous sections, for the comparison of GC and OF-CEAS measurements post data processing was performed to correct for the RGD non linearity and to bring both instruments on the same calibration scale. Additionally, OF-CEAS concentrations have to be expressed in dry air. Humidity rate was not monitored during the flight but was derived from meterological data (analysis from the European Center for Medium-Range Weather Forecasts -ECMWF). The corresponding water vapor obtained vary typically from 0.2% to 1% at respectively high and low altitude inducing a correction on CO values between +0.2 to +1.6 ppb. The model was compared to the closest meteorological station data provided by a radiosounding of Trappes (48$^o$46′N/ 2$^o$1′E/ 168 m asl). These data are recorded by Meteo-France and are available on the SIRTA website (sir). Humidity rates derived from ECWMF model and Trappes data are in agreement within 20%.

In the bottom of Fig. 3 is plotted the difference between OF-CEAS and GC CO concentrations. To be consistent with the typical filling duration of the flasks, OF-CEAS values are averaged during 1 min around the flask filling times. A good agreement is obtained: for the set of ten measurements recorded at different altitudes, the difference has a mean value of -2.2 ppb with a standard deviation of 1.7 ppb. This small systematic difference could not been explained even when different effects like residual effect of the non-linearity of the RGD or humidity correction of the OF-CEAS measurements were examined. The agreement between the OF-CEAS spectrometer and the GC measurements is very close to the 2015 World Meteorological Organization (WMO) compatibility goal at 1 $\sigma$ for CO that is $\pm$ 2 ppb (see WMO/GMA report edited by Tans and Zellweger (2016)).

## 4   Conclusions

The OF-CEAS technique allows for the development of sensitive, compact, robust and reliable instruments to perform in-situ trace gas analysis. After a single calibration with a reference standard, an OF-CEAS instrument delivers in real time absolute CO concentrations that are in excellent agreement with state of the art gas chromatography over one week. Similar performance is expected on other trace molecules for which sufficiently strong absorption lines are available. To reach the best accuracy, the GC is periodically calibrated with a standard gas every 30 min and is corrected from the RGD non linearity with data post processing. The agreement between the OF-CEAS spectrometer and the GC for CO concentrations is typically better than 2 ppb that meets the 2015 WMO recommendation for CO inter-laboratory comparison (Tans and Zellweger, 2016).

OF-CEAS instruments offer other advantages that are rarely associated with high sensitivity and selectivity in gas analysis. The sample volume inside the cavity is below 20 cm$^3$ (standard temperature and pressure) and the pressure can be lowered down to a few mbar, opening field applications in trace detection where small volume samples are available such as bubbles of gas trapped in ice cores for climate studies (Faïn et al., 2014; Rhodes et al., 2016; Grilli et al., 2014). Additionally, the short response time of typically 1 s associated with a high sensitivity (below 1 ppb for CO) can be exploited in different applications such as in breath analysis to distinguish the respiratory phases (Ventrillard-Courtillot et al., 2009; Maignan et al., 2014) or during tropospheric and stratospheric airborne campaigns to deliver high spatial resolution data for atmospheric





models (Romanini et al., 2006; Iannone et al., 2009a). OF-CEAS gas analyzers are now commercialized by AP2E (ProCEAS) that offers presently to measure the concentration of 15 molecular species at high sensitivity with high selectivity.

In order to further enhance the development of the OF-CEAS technique in trace detection and isotopic ratio measurements, the spectral regions that can be exploited have been enlarged to allow new specific molecular absorption signatures.

5    It has been demonstrated that this technique is compatible with different kinds of semiconductor lasers. Indeed, while OF-CEAS was previously developed in the near infrared region with distributed feed-back telecom diode lasers (Morville et al., 2005; Kassi et al., 2006), it has been demonstrated that it is compatible with extended cavity diode lasers that operates in the visible (Courtillot et al., 2006; Horstjann et al., 2014) and with quantum cascade lasers (QCL) (Maisons et al., 2010; Gorrotxategi-Carbajo et al., 2013) and more recently with interband cascade lasers (ICL) (Manfred et al., 2015; Richard et al.,

10    2016) in the mid infrared region.

*Acknowledgements.*  The authors are grateful to the whole RAMCES team for its participation to the flights. This work was partly funded by the CarboEurope-IP EU project and supported by the LabexOSUG@2020 program.



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





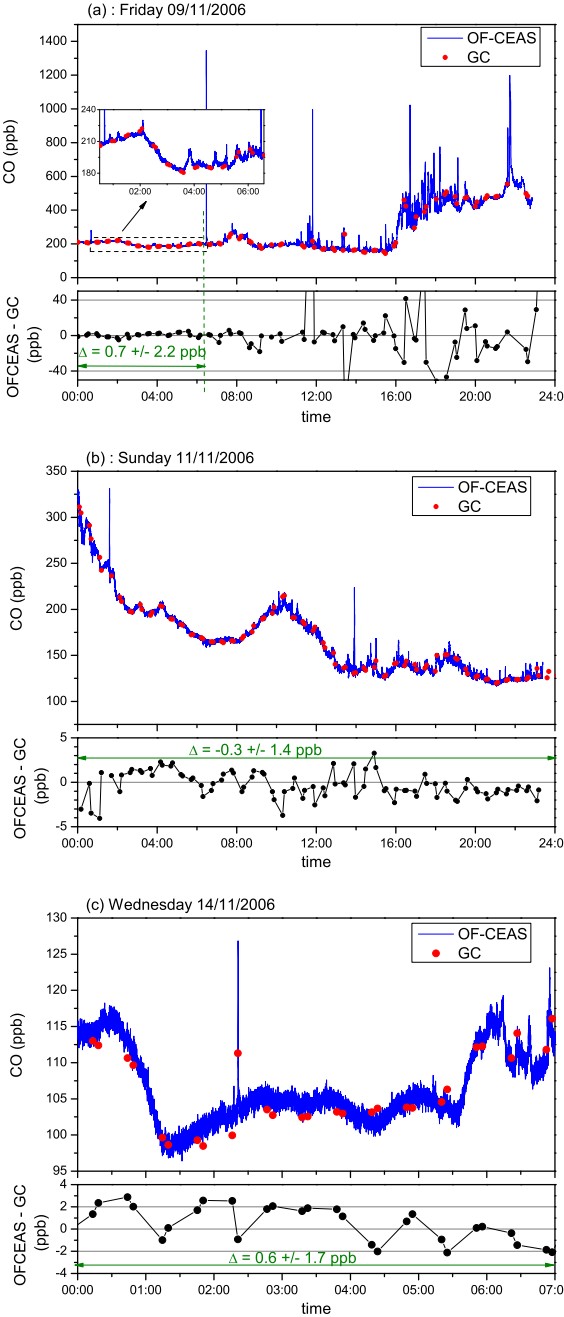

**Figure 2.** Two days and one night monitoring of atmospheric concentration in Gif sur Yvette selected for their different ranges. CO values are given in dry air mole fraction. The GC data are time-shifted by 14 min in order to eliminate time delay between the two instruments. Upper graphs: OF-CEAS measurements are averaged for 2 s while GC measurements are performed twice an hour. Lower graphs: Difference of the measurements after averaging OF-CEAS data for 1 minute around the GC time measurement. Mean values of the difference and standard deviations are written in green.



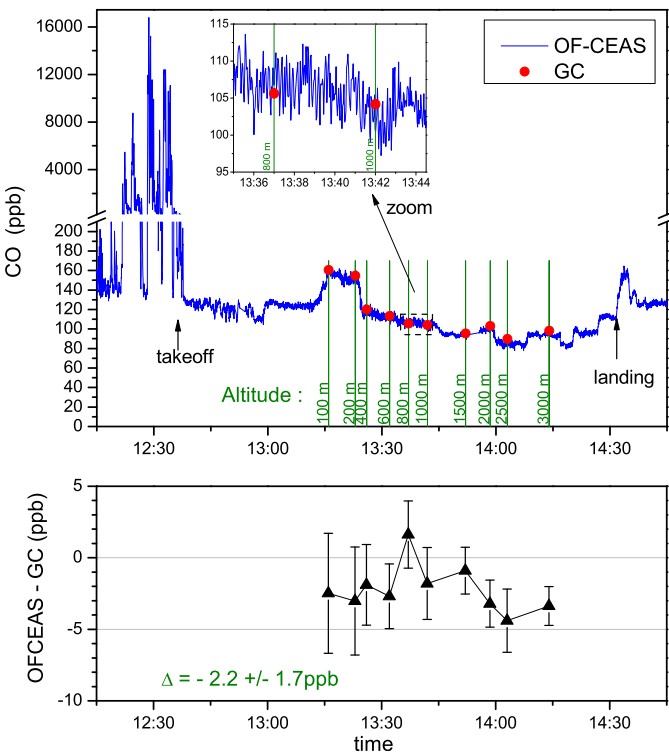

**Figure 3.** Top : CO airborne measurements by OF-CEAS in real time and by GC with latter flask analysis at the LSCE. Altitudes are indicated during collection sample for GC measurements. OF-CEAS data are averaged for 2 s. Bottom: Difference between OF-CEAS and GC measurements, where OF-CEAS values are computed in dry air and averaged for 1 min around the flask filling times. Error bars in this graph indicate the standard deviation of OF-CEAS measurements during 1 min around the comparison time. The difference of OF-CEAS and GC measurements has a mean value of -2.2 ppb with a standard deviation of 1.7 ppb.