# Peer review of "Comparison of cavity enhanced optical–feedback laser spectroscopy and gas chromatography for ground-based and airborne measurements of atmospheric CO concentration"

_Atmospheric Measurement Techniques, 2016_

## Referee Comment (RC1) · Anonymous Referee #1 · 28 Dec 2016

The paper "Comparison of cavity enhanced optical–feedback laser spectroscopy and gas chromatograpy for ground-based and airborne measurements of atmospheric CO concentration", by I. Ventrillard et al., describes a comparison between an old and well known technique, namely gas-chromatograpy, and OF-CEAS, when atmospheric CO is the target molecule, both on the ground and on board a small aircraft.

The work is based on data acquired several years ago, which had to be treated in order to be fully comparable, with recently developed methods.

The paper is clear and well written. This reviewer advices publication of the paper,

once a few points are improved and some typos are corrected.

Page 2, line 10: though based on an "old" spectroscopical technique, an in-situ diode laser based CO analyzer has been deployed on board of the Geophysica aircraft since 2005, with performances comparable to those of the described device: S. Viciani, F. D'Amato, P. Mazzinghi, F. Castagnoli, G. Toci, P.W. Werle: "A cryogenically operated laser diode spectrometer for airborne measurement of stratospheric trace gases", Appl. Phys. B 90, pp. 581-592 (2008). Moreover, analyzers by different firms, (Aerodyne, for instance), use direct absorption in the middle infrared, as very often in this spectral region, and at the target concentrations, few tens of meters are sufficient for measurements at the same level of LOD, resolution and accuracy of the submitted paper. In principle, a good advantage of OF-CEAS, with respect to the above work, is the possibility of using lasers emitting closer to the near infrared, despite the weaker absorption bands. In this wavelength region all the components are generally more user-friendly (and cheaper) than in the middle infrared. Yet, in page 10, lines 5-10, the authors claim (correctly) that any kind of laser (including QCL and ICL, both in the middle infrared) can fit this technique. This reviewer would appreciate a short, further discussion about the motivation for the use of OF-CEAS, in order to provide a clearer picture of the field of application of this technique.

Page 5, line 11: it would be useful to show here Fig. 1 of Kassi et al. (2006), as many readers probably would not go and check that reference, and could ask themselves how to fit a 50 cm cavity (plus some optics) in a 48.26 cm wide rack.

Page 7, line 8: the volume of a 3/8" pipe, 20 m long, is about 1400 cm3. With a flow of "250 sccm" it would take more than 5 minutes to cross the entire pipe length. Could the authors explain their statement?

Typos

Page 3, line 22: "to obtained" should be "to obtain"

Page 5, lens 31 and 32: if ProCEAS is the name of a device, we should have "works" and "reaches"

Page 7, line 3 must be properly formatted

Page 7, line 6: "idenpendent" should be corrected

Page 7, line 23: "close" can be omitted

---

## Author Comment (AC1) · 12 Jan 2017

We are grateful to Referee#1 for his/her careful reading and the helpful criticisms to our manuscript, which will help improving it.

Besides revisiting the manuscript with several minor corrections and reformulations, we list below our responses to the individual issues raised by the reviewer.

Attached is a pdf file of the revised version of the manuscript.

1) Page 2, line 10: though based on an "old" spectroscopical technique, an in-situ diode

laser based CO analyzer has been deployed on board of the Geophysica aircraft since 2005, with performances comparable to those of the described device: S. Viciani, F. D'Amato, P. Mazzinghi, F. Castagnoli, G. Toci, P.W. Werle: "A cryogenically operated laser diode spectrometer for airborne measurement of stratospheric trace gases", Appl. Phys. B 90, pp. 581-592 (2008). Moreover, analyzers by different firms, (Aerodyne, for instance), use direct absorption in the middle infrared, as very often in this spectral region, and at the target concentrations, few tens of meters are sufficient for measure- ments at the same level of LOD, resolution and accuracy of the submitted paper. In principle, a good advantage of OF-CEAS, with respect to the above work, is the possi- bility of using lasers emitting closer to the near infrared, despite the weaker absorption bands. In this wavelength region all the components are generally more user-friendly (and cheaper) than in the middle infrared. Yet, in page 10, lines 5-10, the authors claim (correctly) that any kind of laser (including QCL and ICL, both in the middle infrared) can fit this technique. This reviewer would appreciate a short, further discussion about the motivation for the use of OF-CEAS, in order to provide a clearer picture of the field of application of this technique.

In the introduction section we now more clearly write that the aim of the paper is to compare OFCEAS measurements to the well established GC technique in order to demonstrate for potential new users that this technique is reliable. This is essential to extend the use of OFCEAS beyond the spectroscopist community, for example to atmospheric chemistry, geophysics or medicine. In particular it is relevant to note that OFCEAS instruments are now commercially available (we move this point to the introduction section while it was mentioned at the end of the OFCEAS section, page 5). The comparison of the OFCEAS technique with the numerous other spectroscopy techniques is beyond the scope of this paper. But as advised by the referee, in the introduction we now underline the advantages of the OFCEAS technique. We compare OFCEAS CO analyzers (that are commercially available) to different commercial instruments: we moved here references to Picarro and Los Gatos instruments (previously at the end of the OFCEAS section p5) and added Aerodyne. These instruments

operate in the MIR to reach a sub-ppb LOD. This lead us to present the discussion on the NIR or MIR spectral region in the introduction part.

The introduction part was modified (parts underlined) : Page 2 line 11- page 3 line 6

Among them, Optical–Feedback Cavity Enhanced Absorption Spectroscopy (OF-CEAS) (Morville et al., 2005) exploits a high finesse optical cavity in which is coupled a laser source to enhance the interaction of photons with gas molecules present inside the cavity (Morville et al., 2014). OF-CEAS offers many advantages for quantitative and selective trace gas analysis: it allows real-time absolute measurements with a smallest detectable absorption coefficient in the range of a few 10-10/cm for 1 s acquisition time (Landsberg et al., 2014), it does not require calibration with certified gas mixtures, its sampling volume is small (20 cm3), its response time can be faster than 1 s, and it enables the development of compact instruments to be operated by non-specialists.

Another advantage that follows from the high sensitivity of the OF-CEAS technique is the ability to work in the near infrared region (NIR) where widely used optics are commercially available together with room temperature lasers and detectors. Traditional near infrared (NIR) OF-CEAS instruments reach limit of detection (LOD) at the sub-ppb level for CO(Faïn et al., 2014) that is comparable to other instruments exploiting the mid infrared (MIR) spectral region where absorption coefficient are typically two orders of magnitude higher. Indeed, commercial MIR laser spectrometers based on different laser spectroscopy techniques offer CO sub ppb LOD, such as Picarro instruments by CRDS with a resonant cavity or instruments exploiting a multipass cell like Aerodyne and Los Gatos products. The performance of the OF-CEAS technique in the NIR led a private company (AP2E, Aix-en-Provence, France) to exploit the patent for commercially available analyzers (namely ProCEAS). Exploiting the MIR with OF-CEAS instruments allows to reach sub-ppb levels for several species of interest in trace detection and ppm levels for isotopic ratio measurements (Maisons et al., 2010; Gorrotxategi-Carbajo et al., 2013; Manfred et al., 2016; Richard et al., 2016).

OF-CEAS based measurements of CO have been conducted before around various applications, for example for in-situ trace measurements on geothermal gases (Kassi et al., 2006), for continuous and high resolution measurement of air extracted from ice cores drilled out of polar glaciers (Faïn et al., 2014) or for breath analysis in different medical settings (Ventrillard-Courtillot et al., 2009; Maignan et al., 2014). ProCEAS analyzers are now commercialized in the domains of industrial and air quality monitoring, with some very stringent applications such as air quality control onboard nuclear submarines. In order to further establish for different user communities that OF-CEAS can become a work-horse in many CO applications, which demand robust and compact instrumentation with ppb sensitivity and a fast response time, this paper reports on the comparison of CO measurements performed by OF-CEAS against those obtained by the well established gas chromatography technique. GC measurements were done with a high performance gas chromatograph equipped with a mercuric oxide reduction gas detector (Yver et al., 2009). First, the atmospheric CO concentration in Gif-sur-Yvette, France, was continuously analyzed at ground level over one week. Then, the OF-CEAS instrument was set aboard a small aircraft employed for periodic tropospheric air measurements over the French Orléans forest area. Airborne in-situ CO measurements by OF-CEAS were then compared with flask samples later analyzed with the GC at LSCE.

One sentence at the end of the conclusion section was removed to avoid repetition (previously p2 line 22-25) : With this comparison we demonstrate that OF-CEAS can become a work-horse in many CO applications for environmental (including atmospheric) applications, which demand robust and compact instrumentation with ppb sensitivity and a response time faster than 1 s

2) Page 5, line 11: it would be useful to show here Fig. 1 of Kassi et al. (2006), as many readers probably would not go and check that reference, and could ask themselves how to fit a 50 cm cavity (plus some optics) in a 48.26 cm wide rack. Since 2006, the mechanics has been improved. We prefer not to show again the set-up used for the

measurements reported here and previously published in Kassi et al 2006. We added a mention to the geometry inside the 19" rack : Now page5, line 5-6 : ...fit inside a 19" chassis where the V-shaped cavity is placed in the diagonal as shown in figure 1 of Kassi et al. (2006)

3) Page 7, line 8: the volume of a 3/8" pipe, 20 m long, is about 1400 cm3. With a flow of "250 sccm" it would take more than 5 minutes to cross the entire pipe length. Could the authors explain their statement? It was a mistake, we replaced " about 1 min" by " about 6 min". We clarify the point that the delay between the two instruments (14min deduced from measurements as shown in Figure 2) is mainly due to the cold trap of the GC.

now page 7 Line 15-19 : The estimated sample propagation delay along the tube from the roof to the OF-CEAS instrument is about 6 min (with a gas flow of 250 sccm). A larger delay is observed on the GC data due to the use of the cold trap. The volume of this trap corresponds to the sample volume collected over about 15 min by the GC, inducing a smoothing of the signal of the semicontinuous injections. To elimi-nate the time delay between both instruments, the time shift wasfixed to 14 min (Fig. 2).

Please also note the supplement to this comment:
http://www.atmos-meas-tech-discuss.net/amt-2016-386/amt-2016-386-AC1-supplement.pdf

**Supplement:**

[revised manuscript text omitted]

---

## Referee Comment (RC2) · Anonymous Referee #2 · 28 Feb 2017

Ventrillard et al. presented the comparison results of CO measurements that were made on the ground and on the aircraft using a cavity enhanced optical-feedback laser spectrometer and a GC. The comparison results are interesting; however, as almost all development of the OF-CEAS instrumentation has been published previously, the technical aspect of the manuscript is quite thin. The performance shown in the manuscript is excellent, but it is hard for a reader to judge because the needed information is often not provided. The manuscript will need a major revision to be considered for publication at AMT.

[Figure]

General comments:

1. The detailed setup of the OF-CEAS is missing, especially for the deployment on the aircraft. How was the flow rate controlled? It is mentioned somewhere else that no calibration gas was provided during flight. This could be better shown with a flow diagram.

2. How was humidity correction made to derive dry mole fractions of CO? It was simply mentioned that the effect is small, but without any supporting evidence.

Detailed comments:

P2/L12: change "in which is coupled a laser source" to "in which a laser source is coupled" P3/L10: change "during two steps" to "in two steps" P3/L10: specify that "cooling trap" is the cryogenic cooler P5/L7-10: what is the cavity temperature? Please mention that the response time means 1/e exchange time. P5/L28-35: These are not scientific. Remove them all or write it in a scientific way with proper references. P6/L7-8: what assertion? P6/L17-18: Please elaborate on what has caused the 10% overestimation? As line intensity is well defined within 1% in the HITRAN database. The relative uncertainties on temperature and pressure are also small terms. What else? P6/L32: How is the reproducibility derived? Note that this is often larger than the minimum values derived from the Allan variance. P7/L11-13: How is humidity rate accounted for? Details are needed to judge whether it is properly done. P8/L22: what is the typical vertical distance? P8/L27: you mean "surface", instead of "soilborne", right? P8/32: change "flux to" to "flow of", how is constant flow is maintained? P8/L34: why averaged to 2 second? Note that the response time is much larger now as the flow rate is only 50 sccm. P9/L14: change "been" to "be" P9/L28: what is the exact cavity volume? 20 cm3 or 18 cm3 on P5/L7 P9/L31: it makes no sense to mention the response time when flow rate information is not given. P10/L7: compatible or comparable?

---

## Author Response (AR1)

We are grateful to Referee#2 for his/her careful reading and the helpful criticisms to our manuscript, which will certainly improve it.

As noticed by the referee, the scope of the paper is the comparison of two instruments: a GC and an OF-CEAS analyzer. This is why developments on the OF-CEAS instrumentation was published elsewhere, as well as the work on the GC calibration procedure. Nonetheless, we have modified the manuscript in order to provide some more detailed explanations on data acquisition and analysis.

Besides revisiting the manuscript with several minor corrections and reformulations, we list below our responses (in black) to the individual issues raised by the reviewer (in blue). Changes in the manuscript are underlined.

1) The detailed setup of the OF-CEAS is missing, especially for the deployment on the aircraft. How was the flow rate controlled? It is mentioned somewhere else that no calibration gas was provided during flight. This could be better shown with a flow diagram.

a) To obtain accurate concentration measurements from OFCEAS spectra, the pressure has to be well defined but the flux has no effect (it has an effect on the response time). It is why the pressure is actively controlled while the flux is adjusted manually with a valve. This latest point was indicated in the section on airborne measurements (previously : P8, L31-32 : "*...adjusting a needle valve at the inlet of the sampling cell specifically plumbed to maintain a constant flux to 50 sccm...")* and is now in the description section (2.2). Here, we now clearly indicate that the pressure is actively regulated. According to another comment of Referee2, we clarify the discussion concerning the gas exchange time that depends on the flux value. And we added here the two different values of the flow used in this work, specified before in the two measurements sections (p7, L8) and (p8, L32).

Modifications in the text P5, L2-6 : *The gas is continuously flowing and the cell pressure is stabilized with a pressure regulator, to 200 mbar in this study. The flow is adjusted manually with a needle valve at the inlet of the sampling cell, to 250 sccm (standard cm cubes per minute) for ground measurements and around 50 sccm for airborne measurements. The gas exchange times is then 0.9 s and 4.3 s respectively. If needed, shorter response time can be obtained by using a lower sample pressure or a higher pumping rate.*

b) In this paper we insist that the OFCEAS instrument has to be calibrated only once to deliver absolute concentration measurements and then run without calibration gas. In the introduction section we now list the advantages of this technique, and among them "*it does not require periodic calibrations with certified gas mixtures*" (p2, L15).

Then, this point is discussed in section 2.2 (P5, L9-10) :"*Importantly, OF-CEAS provides quantitative absorption measurements in real-time without the need for a periodic calibration with certified gas mixtures...*", later in detail in the sub-section "*Calibration of the OF-CEAS Spectrometer: Conversion of absolute molecular absorption to CO concentration*" and mentioned again in the conclusion (P10 L19) "*After a single calibration with a reference standard, an OF-CEAS instrument delivers in real time absolute CO concentrations*".

2) How was humidity correction made to derive dry mole fractions of CO? It was simply mentioned that the effect is small, but without any supporting evidence.
As the humidity rate was not directly monitored, we made the following correction :

a) For ground measurements dry fraction is derived from the humidity rate and atmospheric temperature and pressure measured at the Saclay tower located 1.5km away from the sampling point. We detailed the calculation P7 L29-36

*Reported GC CO values are dry air mole fractions. For the comparison, OF-CEAS CO mixing ratio (xCO,air) is converted into dry air mixing ratio (xCO,dry) according to:*
*xCO,dry =xCO,air/(1-xH2O )= xCO,air/(1-[RH x e(T)]/P)*
*Where the water mixing ratio (xH2O) is computed from the humidity rate (RH), the atmospheric pressure (P) and the water saturation vapor pressure (e(T)) given by a polynomial function of the atmospheric temperature (T) (Lowe, 1976). The meteorological*
*data used (RH, P and T) are routinely monitored at the Saclay tower* *located about 1.5 km North-Northwest from the sampling point (data provided by the SPR group from Saclay CEA).*

b) For airborne measurements, we could compare two different corrections : P9, L32:

*OF-CEAS concentrations have to be expressed in dry air. The water mixing ratio was not monitored during the flight but was derived from a model allowing to compute the specific humidity q from meteorological data (analysis from the European Center for Medium-Range Weather Forecasts - ECMWF). Water mixing ratio is then given by:*
*xH2O = q.Mdry/(MH2O +q.(Mdry -MH2O))*
 *Where MH2O and Mdry are respectively the molar mass of water and dry air.* *During the flight, the values obtained for the water mixing ratio vary typically from 0.2% to 1% at respectively high and low altitude inducing a correction on CO values between +0.2 to +1.6 ppb. The model was compared to the meteorological data (RH, P and T) provided by the closest station that is the radiosounding of Trappes (48o46'N/ 2o1'E/ 168m asl). These data are recorded by Meteo-France and are available on the SIRTA website (sirta). Humidity rates derived from ECWMF model and Trappes data are in agreement within 20%, which means that corrections obtained from one or the other model will be closer than the measurement error.*

Detailed comments:
- P5/L7-10: what is the cavity temperature?

We added P5, L8 : *The device is temperature stabilized around 22ºC using adhesive heating ribbons.*

- Please mention that the response time means 1/e exchange time.

modification P4, L33 : "*However the measurement response time for 1/e change in a concentration value is not limited by this rate but by the gas exchange rate...*"

- P5/L28-35: These are not scientific. Remove them all or write it in a scientific way with proper references.

References to commercial instruments was moved to the introduction section (see response to the first comment of referee1 ).
P2, L22-25 : *Indeed, commercial MIR laser spectrometers based on different laser spectroscopy techniques offer CO sub ppb LOD, such as Picarro instruments by CRDS with a resonant cavity or instruments exploiting a multipass cell like Aerodyne and Los Gatos products. The performance of the OF-CEAS technique in the NIR led a private company (AP2E, Aix-en-Provence, France) to exploit the patent for commercially available analyzers (namely ProCEAS).*

- P6/L17-18: Please elaborate on what has caused the 10% overestimation? As line intensity is well defined within 1% in the HITRAN database. The relative uncertainties on temperature and pressure are also small terms. What else?

Indeed temperature and pressure uncertainties cannot explained this shift that would corresponds to an error on the temperature of tens of degrees or an error on the pressure of tens of mbar. Another bias that can be neglected is the interference with other trace species present in the standards: only CH4 can interfere in the spectral region scanned by the instrument. But CH4 concentration is simultaneously measured by the OF-CEAS instrument.

We think that the 10% difference with HITRAN database is mainly due to the standard calibration. As explained in section 2.2, we initially used standards certified by the CSIRO to calibrate the instrument, then later comparing CSIRO and WMO (from NOAA) standards (by using another OFCEAS device). We reevaluated the two CSIRO standards on the WMO scale and found the second scale displayed 5% and 7% deviations relative to the first. In order to conclude our comparison with the chromatographic measurements we had to translate our measurements from CSIRO to NOAA scale, which then makes our measurements about 10% larger when directly using HITRAN absorption line intensities (calculated at the instrument T and P. Thus, we then believe that the overestimation by 10% when using HITRAN line intensities is not an instrumental problem but a problem with the chosen calibration standards. It is clear that the difference between the two calibration scale is not consistent with HITRAN based measurements at the 1% level. To make the argument short and not too much polemic we added the following lines (P7, L6-11) :

*The accuracy of this calibration is estimated to be of 2% limited by the accuracy of the NOAA standards. But the obtained conversion factor corresponds to a 10% overestimation of the line intensity specified in HITRAN with a 1% accuracy. CSIRO specifications of the two standards being offset by 5% and 7% as compare to NOAA standards are neither compatible with HITRAN database. Nonetheless, the good agreement of OF-CEAS and GC measurements reported in the following shows that this calibration on the same reference scale is a crucial point for the inter-comparison.*

and we added in the conclusion (P10, L19) :
*After a single calibration with a reference standard, an OF-CEAS instrument delivers in real time absolute CO concentrations that are in excellent agreement over one week with state of the art gas chromatograph referenced to the same calibration scale.*

- P6/L32: How is the reproducibility derived? Note that this is often larger than the minimum values derived from the Allan variance.

The number of 0.2ppb in 20s refers to Allan variance measurements. We changed "reproducibility" into " detection limit". In the previous version of the manuscript we mentioned long term drift in P5, L23-25. It is now more detailed in P7, L12-17 :

*The LOD of the OF-CEAS spectrometer is much smaller, at the level of 0.2 ppb for an acquisition time of 20 s. At longer acquisition times, small drifts prevent for better averaging. It is partly to be attributed to drifts in the sensors that are used to control sample pressure and temperature, thus selection of more stable sensors can decrease the drift. Other causes of drift are changes in parasitic optical etalon effects (Morville et al., 2014). However the drifts associated to these optical effects can be made quite small and cannot increase arbitrarily and remain bounded at all times as shown by the Allan variance of CO measurements in Faïn et al. (2014).*

And in the conclusion (P10, L14-18) : *The agreement between the OF-CEAS spectrometer and the GC for CO concentrations is typically better than 2 ppb that meets the 2015 WMO recommendation for CO inter-laboratory comparison (Tans and Zellweger, 2016). This agreement shows that OF-CEAS instrumental drift on the long term remains acceptable, at the level of accuracy required for atmospheric CO monitoring. Periodic calibrations with a standard gas could become necessary to attain a higher degree of accuracy, since these calibrations could be used to correct the effect of these small drifts.*

- P7/L11-13: How is humidity rate accounted for? Details are needed to judge whether it is properly done.

See response to the second comment.

- P8/L22: what is the typical vertical distance?

P9 : L3/4 : *Glass flasks are filled at 10 different altitudes (between 100m and 3000m above ground level).*

- P8/32: how is constant flow is maintained?

In relation to the first comment response. During airborne measurement, the ambient pressure was subject to significant changes, requiring the adjustment of the manual valve. It is now detailed in this section (P9, L21-24) :

*The only not automatized action by the operator required during the flight consisted in adjusting the flow in the sampling cell with a needle valve. Given that this valve was placed at instrument inlet, the flow changed linearly with pressure, thus decreased with altitude. However for GC comparison, measurements were taken during constant altitude sections, allowing averaging on time scales largely exceeding the sample exchange time for flow around 50 sccm.*

- P8/L34:why averaged to 2 second? Note that the response time is much larger now as the flow rate is only 50 sccm.

Data are now averaged to 5.5s that corresponds to the largest response time value during the flight (for a flow of 40sccm) => Top part of Figure 3 was modified.
Modifications in the text P9, L25: *During the flight, the flow was slowly varying between 40 sccm and 70 sccm. OF-CEAS data are averaged for 5.5 s to be consistent with the largest value of the response time.*

- P9/L28: what is the exact cavity volume? 20 cm3 or 18 cm3 on P5/L7

The sample cavity is 18cm3 for this specific instrument. It could be smaller in other OF-CEAS instruments. It is why in the conclusion section we have written more generally "*below 20cm3*".

- P9/L31: it makes no sense to mention the response time when flow rate information is not given.

Indeed as discussed in the reply to the first comment, the response time depends on the flow and the pressure. In the concluding remarks we just want to underline that OFCEAS instruments can be developed to reach small response time if needed.

[revised manuscript text omitted]

---

## Author Response (AR2)

Referee#2 :

Thanks for your efforts in addressing my comments Referee#2. I am in general satisfied with your responses, with the following few technical corrections, according to the revised version with track changes:

1) What type of pressure regulator was used to stabilize the cell pressure? Is the needle valve upstream or downstream of the pressure regulator?

The needle valve is upstream ("at the inlet of the sampling cell" , L3 P5), the pressure regulator is downstream . This latest point is now in L2 P5 :

*The gas is continuously flowing and the cell pressure is stabilized with a downstream pressure regulator, to 200 mbar in this study. The flow is adjusted manually with a needle valve at the inlet of the sampling cell,....*

2) The conversion of ECMWF specific humidity: it should be $H_2O$ mole fraction, instead of mixing ratio, otherwise the equation you use for $xH_2O$ and for water correction will not be correct.

We replaced all occurrences of *mixing ratio* by *mole fraction*

3) "flux" was used at several places, and to my understanding should be "flow"

corrected